# Enhancing 1,3-Propanediol Productivity in the Non-Model Chassis *Clostridium beijerinckii* through Genetic Manipulation

**DOI:** 10.3390/microorganisms11071855

**Published:** 2023-07-22

**Authors:** Jonatã Bortolucci, María-Eugenia Guazzaroni, Teresa Schoch, Peter Dürre, Valeria Reginatto

**Affiliations:** 1Departamento de Química, Faculdade de Filosofia, Ciências e Letras de Ribeirão Preto, Universidade de São Paulo, Av. Bandeirantes, 3900, Ribeirão Preto 14040-030, SP, Brazil; jonata.bortolucci@usp.br; 2Departamento de Biologia, Faculdade de Filosofia, Ciências e Letras de Ribeirão Preto, Universidade de São Paulo, Av. Bandeirantes, 3900, Ribeirão Preto 14040-030, SP, Brazil; meguazzaroni@ffclrp.usp.br; 3Institut für Mikrobiologie und Biotechnologie, Universität Ulm, Albert-Einstein-Allee, 11, D-89081 Ulm, Germany

**Keywords:** glycerol dehydratase, 1,3-propanediol dehydrogenase, non-solventogenic, modular vector, non-model chassis, *Clostridium*, transformation

## Abstract

Biotechnological processes at biorefineries are considered one of the most attractive alternatives for valorizing biomasses by converting them into bioproducts, biofuels, and bioenergy. For example, biodiesel can be obtained from oils and grease but generates glycerol as a byproduct. Glycerol recycling has been studied in several bioprocesses, with one of them being its conversion to 1,3-propanediol (1,3-PDO) by *Clostridium*. *Clostridium beijerinckii* is particularly interesting because it can produce a range of industrially relevant chemicals, including solvents and organic acids, and it is non-pathogenic. However, while *Clostridium* species have many potential advantages as chassis for synthetic biology applications, there are significant limitations when considering their use, such as their limited genetic tools, slow growth rate, and oxygen sensitivity. In this work, we carried out the overexpression of the genes involved in the synthesis of 1,3-PDO in *C. beijerinckii* Br21, which allowed us to increase the 1,3-PDO productivity in this strain. Thus, this study contributed to a better understanding of the metabolic pathways of glycerol conversion to 1,3-PDO by a *C. beijerinckii* isolate. Also, it made it possible to establish a transformation method of a modular vector in this strain, therefore expanding the limited genetic tools available for this bacterium, which is highly relevant in biotechnological applications.

## 1. Introduction

The bioeconomy, also referred to as the biomass-based economy, aims to replace the current economic model with a more sustainable one based on renewable resources [1]. A key strategy in the bioeconomy is the use of biorefineries, which are industrial units that integrate biomass conversion processes to produce biofuels, power, and chemicals from biomass [2].

Besides lignocellulosic biomasses, algae and various types of industrial byproducts and waste can be used in different biorefineries installations [3]. Glycerol, for example, generated from the transesterification reaction in the biodiesel manufacture, accounts for approximately 10% (*w*/*w*) of the total biodiesel produced [4]. Exceeding market demand, the annual production of crude glycerol reached 7.66 million tons, with an estimated annual increase of 3.5% by 2025 [5,6]. Chemical approaches have helped to purify the residual glycerol from biodiesel, but excess glycerol can be used as a carbon source in fermentative processes [7]. Since glycerol is a more reduced substrate compared to glucose or xylose, normally employed in biorefineries, it yields double the amount of reducing equivalents [8]. Thus, glycerol has been considered an attractive substrate for obtaining products from reductive fermentation routes, such as 1,3-propanediol (1,3-PDO). 1,3-PDO is a reduced bioproduct that can be obtained via the biological conversion of glycerol, which is mostly employed in the manufacture of various polymers, such as polyethers, polyurethanes, and polyesters [9,10].

Bacteria, especially those belonging to the *Klebsiella*, *Citrobacter*, *Enterobacter*, *Lactobacillus*, and *Clostridium* genera, have the ability to metabolize glycerol into 1,3-PDO [11]. The most commonly studied species for this purpose are *Klebsiella pneumoniae*, *Clostridium butyricum*, and *Clostridium pasteurianum*. However, most *Klebsiella* strains are pathogenic, and, thus, non-pathogenic *Clostridium* strains are preferred due to their good yields [12,13].

Glycerol fermentation by *Clostridium* strains can be represented by the metabolic reactions illustrated in Figure 1, which consists of two different branches [11,14,15]. In the oxidative branch, ATP and reducing equivalents are generated, which lead to cellular growth and maintenance with the synthesis of fermentation products, such as organic acids (acetate and butyrate) and organic solvents (acetone, butanol, and ethanol). Glycerol is first converted to dihydroxyacetone by the enzyme glycerol dehydrogenase (GDH, encoded by *dhaD* gene), which is then phosphorylated to dihydroxyacetone phosphate by dihydroxyacetone kinase (DHAK, encoded by *dhaK* gene).

In the reductive branch, the reducing equivalents are regenerated by first converting glycerol to 3-hydroxypropionaldehyde by coenzyme B_12_-independent glycerol dehydratase (GDHt), which is encoded by *dhaB1* and *dhaB2* genes. This intermediate is then reduced to 1,3-PDO by 1,3-propanediol dehydrogenase (PDODH), which is encoded by the *dhaT* gene.

Among *Clostridium* species, *Clostridium beijerinckii* is a versatile biocatalyst that can produce a wide range of products depending on the substrate, such as organic acids, butanol, and 1,3-PDO from glycerol fermentation [17]. Our research group has been working with an isolate [18], *C. beijerinckii* Br21, which has been studied for its potential to produce hydrogen [18,19], butyric acid [19,20,21,22], and 1,3-propanediol [23,24]. The advantage of using our isolate, *C. beijerinckii* Br21, is that it lacks the *adc* gene that encodes for acetoacetate decarboxylase (Figure 1) [25]. This natural deletion renders the strain incapable of conventional acetone, butanol, and ethanol (ABE) fermentation, normally performed by *C. beijerinckii* [26]. This can be advantageous for the Br21 strain, as it could provide more reducing equivalents (NADH) for the 1,3-PDO reductive branch. When performing glycerol fermentation, *C. beijerinckii* Br21 generates 1,3-PDO (C_3_H_8_O_2_) as the main product, as well as acetic acid (C_2_H_4_O_2_), butyric acid (C_4_H_8_O_2_), and hydrogen (H_2_), according to Equations (1)–(3), respectively [27,28]. Combining these equations [(6 × Equation (1)) + (2 × Equation (2)) + Equation (3)], which represent the synthesis of each individual product, results in the global equation of glycerol fermentation by *Clostridium* (Equation (4)). From this stoichiometry, the maximum yield of 1,3-PDO in glycerol fermentation is 0.6 mol mol^−1^.
C_3_H_8_O_3_ + NADH + H^+^ → C_3_H_8_O_2_ + NAD^+^ + H_2_O(1)
C_3_H_8_O_3_ + 2 NAD^+^ + H_2_O → C_2_H_4_O_2_ + H_2_ + CO_2_ + 2 NADH + 2 H^+^ + 2 ATP(2)
2 C_3_H_8_O_3_ + 2 NAD^+^ → C_4_H_8_O_2_ + 2 H_2_ + 2 CO_2_ + 2 NADH + 2 H^+^ + 3 ATP(3)
10 C_3_H_8_O_3_ → 6 C_3_H_8_O_2_ + 2 C_2_H_4_O_2_ + C_4_H_8_O_2_ + 4 H_2_ + 4 CO_2_ + 7 ATP + 4 H_2_O(4)

A number of synthetic biology tools have been developed to enable metabolic engineering in *Clostridium* [29], which include efficient gene editing and targeted disruption by the ClosTron technology [30,31], allelic coupled exchange [32,33], and CRISPR/Cas9 system [34,35]. Random mutagenesis was also accomplished using transposon mobile elements [36,37]. In addition to the development of these technologies, great attention has been given to the study of different biological parts, such as promoters, reporters, origins of replication, and terminators, which affect the productivity and yield of desired products [29]. However, while *Clostridium* species have many potential advantages as chassis for synthetic biology applications, there are still significant limitations when considering their use, such as limited genetic tools available, restricted transformation protocols, and a small number of biological parts already tested for genetic circuits construction [38]. 

In this study, we conducted gene overexpression experiments of the genetic cluster *dhaB1*, *dhaB2*, *pduO*, and *dhaT* to enhance the synthesis of 1,3-PDO in *C. beijerinckii* Br21. This approach resulted in a 35% increase in the productivity of 1,3-PDO in the modified strain compared to the non-transformed clone (0.27 and 0.20 mmol L^−1^ h^−1^, respectively). Consequently, our research contributed to a deeper understanding of the metabolic pathways involved in the conversion of glycerol to 1,3-PDO by a non-solventogenic *Clostridium beijerinckii* isolate. While other studies have explored the expression of similar genes or related counterparts [9,39], here, a non-solventogenic strain has been manipulated. This characteristic potentially results in higher availability of reducing equivalents for 1,3-PDO synthesis. Additionally, we successfully established for the first time a transformation method for introducing a modular shuttle vector into this strain, thereby expanding the repertoire of genetic tools available for this bacterium.

## 2. Materials and Methods

### 2.1. Media, Strains, and Growth Conditions

*C. beijerinckii* Br21 was isolated from a sugarcane mill wastewater treatment plant [18,25]. The Whole Genome Shotgun project of this strain is deposited at DDBJ/ENA/GenBank and can be accessed through the National Center for Biotechnology Information (NCBI) database, under the access number MWMH00000000. The strain was maintained in Reinforced Clostridial Medium (RCM) (10 g L^−1^ peptone, 10 g L^−1^ meat extract, 3 g L^−1^ yeast extract, 5 g L^−1^ glucose, 5 g L^−1^ NaCl, 1 g L^−1^ soluble starch, 0.5 g L^−1^ cysteine, 3 g L^−1^ sodium acetate and 0.5 g L^−1^ agar) with 30% glycerol, stored at −80 °C.

Starting with 50 µL of the freezer stock (stored at −80 °C) as inoculum, the initial culture was carried out in *Hungate* tubes containing 5 mL of RCM, at 37 °C and without agitation, for 18 h. After this, the media employed varied according to the further experiment. For transformation, 2xYTG medium (16 g L^−1^ tryptone, 10 g L^−1^ yeast extract, 5 g L^−1^ NaCl and 5 g L^−1^ glucose) was used. For the growth assay, 60 mL of WIS medium (5 g L^−1^ K_2_HPO_4_, 0.5 g L^−1^ yeast extract, and 0.005 g L^−1^ sodium acetate) [40] was used in 150 mL flasks (in triplicate). In all cases, anaerobic conditions were accomplished by gas phase exchange using a mixture of 80% N_2_ and 20% CO_2_. For the gas phase exchange, a vacuum was applied to the headspace of the flasks until the pressure reached 60 mbar. This was followed by purging the flasks with the 80:20 mixture of N_2_ and CO_2_ until the pressure reached 1 bar. The vacuum-purge procedure was repeated seven times to ensure the establishment of anaerobic conditions inside the flasks.

For obtainment and replication of the plasmids, a culture of *Escherichia coli* XL1-Blue MRF’ stored at a −80 °C freezer was used. The propagation and replication of this microorganism were carried out in LB medium (10 g L^−1^ tryptone, 5 g L^−1^ yeast extract, and 10 g L^−1^ NaCl), incubated at 37 °C and 200 rpm, for 18 h. 

After transforming *E. coli* XL1-Blue MRF’ and *C. beijerinckii* Br21, the media was supplemented with antibiotics at appropriate concentrations of 250 and 2 µg mL^−1^ of erythromycin and clarithromycin, respectively.

### 2.2. Plasmid Construction, Replication and Sequencing

For the amplification of the DNA fragment containing *dhaB1*, *dhaB2*, *pduO*, and *dhaT* genes (5015 bp) (Figure 2), here referred to as the 1,3-PDO gene cluster, genomic DNA was initially extracted from *C. beijerinckii* Br21 by employing the MasterPure^TM^ Gram Positive DNA Purification Kit (Epicentre Biotechnologies, Madison, WI, USA). The necessary primers for the amplification (Table 1) of the 1,3-PDO gene cluster were designed using NEBuilder^®^ Assembly Tool (https://nebuilder.neb.com, accessed on February 2022) and synthesized by Biomers GmbH (Ulm, Baden-Württemberg, Germany). The PCR was accomplished using CloneAmp HiFi PCR Premix (Clontech, Mountain View, CA, USA) under the following conditions: 98 °C for 10 s, 56 °C for 15 s, and 72 °C for 150 s, repeating these three steps 33 times. A negative control was also performed by replacing the genomic DNA in the reaction with DNA-free water.

The backbone used for construction of the plasmid harboring the 1,3-PDO gene cluster was pMTL83251 (Figure 3a) [41]. The promoter of the phosphotransacetylase-acetate kinase encoding genes from *Clostridium ljungdahlii* was employed as constitutive promotor. Starting from another plasmid, pMTL83251_P*_pta-ack_*_1,3-propanediol_CLOBI (Figure 3b) [39], the backbone containing the promotor (4896 bp) was recovered using FastDigest^®^ *Bsp119*I, *Nhe*I, and *Xba*I (Thermo Fisher Scientific, Waltham, MA, USA) restriction enzymes. Before cloning, both backbone and 1,3-PDO gene cluster fragments were purified using DNA Clean & Concentrator^®^-5 (Zymo Research Corporation, Irvine, CA, USA). Next, the cloning reaction was performed using NEBuilder HiFi DNA Assembly Master Mix (Clontech, Mountain View, CA, USA), for joining the backbone with the 1,3-PDO gene cluster, resulting in pMTL83251_P*_pta-ack_*_1,3-PDO_cluster (Figure 3c).

After constructing the plasmid pMTL83251_P*_pta-ack_*_1,3-PDO_cluster, it was transformed in *E. coli* XL1-Blue MRF’ for replication. After overnight incubation of the culture in LB medium, plasmid DNA was extracted using Zyppy^TM^ Plasmid Miniprep Kit (Zymo Research Corporation, Irvine, CA, USA). To confirm the size of the constructed plasmid pMTL83251_P*_pta-ack_*_1,3-PDO_cluster, plasmid DNA was digested with *Pvu*II and *Xba*I restriction enzymes, which would generate 6445 and 3422 bp size fragments. Furthermore, plasmid DNA was sent for sequencing to Microsynth Seqlab GmbH (Göttingen, Niedersachsen, Germany) to verify the absence of mutations. The sequencing reaction was targeted at the 1,3-PDO gene cluster region of the plasmid using primers (Biomers GmbH, Ulm, Baden-Württemberg, Germany) listed in Table 1.

### 2.3. Transformation of C. beijerinckii Br21 with pMTL83251_P_pta-ack__1,3-PDO_Cluster

Before carrying out the transformation of *C. beijerinckii* Br21, a search was performed throughout the genome to identify potential restriction-modification (RM) systems. Several genes related to a type I RM system (*hsd*) were found, namely *hsdR* (locus tag: CBEIBR21_17865, CBEIBR21_00480, CBEIBR21_13600), which encodes the endonuclease; and *hsdM* (locus tag: CBEIBR21_17880), which encodes the methylase. In type I RM systems, the methylated base formed is m6A (N6-methyladenosine) [42]; conveniently, *E. coli* XL1-Blue MRF’ that was used for construction and replication of the plasmid also encodes a type I RM system, as well as a mutation (*hsd*R17) that eliminates the endonuclease, which hinders its ability to degrade unmethylated DNA. Nevertheless, it still encodes a functional m6A base methylase. Therefore, the pMTL83251_P*_pta-ack_*_1,3-PDO_cluster plasmid replicated by *E. coli* XL1-Blue MRF’ exhibits the correct methylation pattern for transformation in *C. beijerinckii* Br21.

The transformation of *C. beijerinckii* Br21 with pMTL83251_P*_pta-ack_*_1,3-PDO_cluster plasmid was performed according to the protocol described by Little and collaborators [43], which was developed for the transformation of *C. beijerinckii* NCIMB 14988; some adjustments were necessary for the successful transformation of the Br21 strain. All the procedures were done under anaerobic conditions, inside an anaerobic chamber, with a gas atmosphere composed of 95% N_2_ and 5% H_2_. When cells reached the exponential growth phase, this culture was used to inoculate 60 mL of 2xYTG medium on a 150 mL flask, with initial OD_600_ of 0.05. When OD_600_ reached 0.6, the cells were centrifuged at 2000 rpm and 4 °C for 10 min. Then, the cell pellet was washed using decreasing volumes of cold SEB buffer (270 mmol L^−1^ sucrose, 7 mmol L^−1^ Na_2_HPO_4_, 7 mmol L^−1^ NaH_2_PO_4_, and 1 mmol L^−1^ MgCl_2_) [43]: 40 and 20 mL, respectively. Finally, the cell pellet was suspended in 5 mL of cold SEB buffer containing 10% (*v*/*v*) DMSO; a total of 280 µL aliquots of this cell suspension were stored at the −80 °C freezer.

For transformation, 1 µg of plasmid DNA was added to the 280 µL *C. beijerinckii* Br21 cells aliquot, and this solution was transferred to an 0.2 cm electroporation cuvette (Bulldog Bio, Biozym Scientific GmbH, Hessisch Oldendorf, Niedersachsen, Germany). An exponential pulse of 1.25 kV was applied, with 200 Ω resistance and 25 µF capacitance, resulting in a time constant of 3 ms. Immediately after the electrical pulse, cells were recovered in Hungate tubes containing 3 mL of 2xYTG medium. The cultures were incubated at 37 °C for 5 h. After incubation, 500 µL of each culture were transferred to new Hungate tubes containing fresh media, supplemented with 2 µg mL^−1^ clarithromycin. When growth was observed in the presence of antibiotic, 100 µL of the culture was used to inoculate Petri dishes containing 2xYTG with 1.5% (*w*/*v*) agar. After incubation at 37 °C for 2 to 4 days, colonies were picked and grown in liquid RCM, containing 2 µg mL^−1^ clarithromycin.

### 2.4. Confirmation of Transformation

After growth of transformed *C. beijerinckii* Br21 in RCM containing 2 µg mL^−1^ clarithromycin, a verification of the size of the harbored plasmid and identity of the transformed strain was performed. An overnight culture of the transformed strain was used for the total DNA extraction using MasterPure^TM^ Gram Positive DNA Purification Kit (Epicentre Biotechnologies, Madison, WI, USA). Then, total DNA was transformed into *E. coli* XL1-Blue MRF’, for replication of the plasmid. Finally, plasmid DNA was extracted from an overnight culture of transformed *E. coli* XL1-Blue MRF’ using Zyppy^TM^ Plasmid Miniprep Kit (Zymo Research Corporation, Irvine, CA, USA). To confirm the transformation with pMTL83251_P*_pta-ack_*_1,3-PDO_cluster, plasmid DNA was digested with FastDigest^®^ *Pvu*II and *Xba*I restriction enzymes, which would generate 6445 and 3422 bp size fragments.

Moreover, the identity of the transformed strain was checked by amplification and sequencing of 16S rRNA gene. For the amplification, an overnight culture of the transformed *C. beijerinckii* Br21 was used for the genomic DNA extraction as described in Section 2.2. The primers used to amplify the 16S rRNA gene were synthesized by Biomers GmbH (Ulm, Baden-Württemberg, Germany) and are listed in Table 1. Then, PCR was performed using ReproFast-DNA Polymerase (Genaxxon BioScience GmbH, Ulm, Baden-Württemberg, Germany), with the following conditions: 95 °C for 30 s, 55 °C for 30 s, and 72 °C for 90 s, repeating these three steps for 30 times. After confirming amplification of the correct size fragment (1465 bp) by gel electrophoresis using agarose gel (0.8% *w*/*v*) and TAE 1X buffer, the fragment was sent for sequencing to Microsynth Seqlab GmbH (Göttingen, Niedersachsen, Germany). Then, the sequenced fragment was aligned to *C. beijerinckii* Br21 genome, and strain identity was confirmed by similarity above 99%.

### 2.5. Bacterial Growth Experiment

After transformation of *C. beijerinckii* Br21 with pMTL83251_P*_pta-ack_*_1,3-PDO_cluster, a batch mode growth experiment was performed with both the wild and the transformed strains. The medium described by Wischral and collaborators [40] was employed, which is referred to here as WIS medium. It is composed of 5 g L^−1^ K_2_HPO_4_, 0.5 g L^−1^ yeast extract, and 0.005 g L^−1^ sodium acetate. Before autoclaving, pH was adjusted to 7.2. A stock solution of glycerol was separately autoclaved and aseptically added to the culture medium to obtain an initial concentration of 500 mmol L^−1^. This concentration was selected based on a previous work performed by our group [23], in which 500 mmol L^−1^ glycerol as initial carbon source yielded the highest final concentration of 1,3-PDO.

The batch fermentation assay was carried out on 120 mL flasks containing 60 mL of WIS medium. At determined points of the fermentation, samples were collected from the culture medium for measurements of OD_600_ and pH. Furthermore, substrate and product concentrations of samples were also measured: glycerol and 1,3-PDO were analyzed by HPLC; acetate, butyrate, ethanol, and butanol were analyzed by GC.

### 2.6. Substrate and Products Quantification

To perform the analyses, samples were first centrifuged at 15,000 rpm and 4 °C for 35 min. Then, 480 µL of the supernatant was collected and used for the analyses. 

Gas Chromatography (GC) was used to quantify acetate, butyrate, ethanol, and butanol. In this analysis, 20 µL 2 mol L^−1^ HCl was also added to the collected supernatant. The GC analysis was performed using Clarus 600 (PerkinElmer Corp., Waltham, MA, USA) equipped with Elite-FFAP Capillary Column with 30 m length size, 0.35 mm inner diameter, and 0.25 µm film thickness (PerkinElmer Corp., Waltham, MA, USA) containing nitroterephthalic acid modified polyethylene glycol (PEG) polymer as stationary phase. H_2_ was used as carrier gas, with a flow rate of 2.25 mL min^−1^. The detector used for analysis was FID, set to 300 °C. Injection volume was 1 µL with 1:20 split, and injector temperature was set to 225 °C. The column temperature during analysis was varied as follows: 70 °C for 2 min, increasing to 250 °C (ramp rate of 40 °C min^−1^), 250 °C for 1 min. Analysis time was set to 7.5 min.

High-performance liquid chromatography (HPLC) was used to quantify glycerol and 1,3-PDO. It was performed using HPLC 1290 Infinity (Agilent Technologies, Santa Clara, CA, USA) equipped with CS-Organic Acid Resin Column 300 × 8 mm (CS-Chromatographie Service GmbH, Langerwehe, Nordrhein-Westfalen, Germany) containing polystyrene-divinylbenzene-copolymer as stationary phase. The precolumn consisted of the same stationary phase, with 40 × 8 mm size. The column temperature was set to 40 °C. A solution of 5 mmol L^−1^ H_2_SO_4_ was used as mobile phase, with a flow rate of 0.6 mL min^−1^. The detectors used for analysis were RID and DAD/UV (210 nm). Injection volume was 20 µL, and analysis time was set to 30 min.

### 2.7. Kinetic Parameters Determination

The results from the bacterial growth experiment were used to calculate the specific rates of cell growth, substrate consumption, and product formation.

In order to calculate the specific growth rate (µ), the OD_600_ values (z) were converted to dry cell mass concentration (X) (mg L^−1^), using a correlation that was determined experimentally (Equation (5)).
X = 0.0024 · z(5)

A sigmoidal fitting was performed on the scatter plot of dry cell mass as a function of time [X = f(t)], using the Boltzmann equation. µ (h^−1^) was determined by the first derivative of X = f(t), divided by X, as shown in Equation (6).
(6)$$\mu =\left(\frac{dX}{dt}\right)\cdot \left(\frac{1}{X}\right)$$

The specific glycerol consumption rate (q_Glycerol_) (mmol glycerol mg^−1^ dry cell mass h^−1^) and the specific 1,3-PDO formation rate (q_1,3-PDO_) (mmol 1,3-PDO mg^−1^ dry cell mass h^−1^) were calculated using the same procedure, based on their respective concentrations as a function of time scatter plots (Equations (7) and (8)).
(7)$$q_{Glicerol}=\left(\frac{dGlicerol}{dt}\right)\cdot \left(\frac{1}{X}\right)$$
(8)$$q_{1,3-PDO}=\left(\frac{d1,3-PDO}{dt}\right)\cdot \left(\frac{1}{X}\right)$$

### 2.8. Non-Dissociated Butyric Acid Concentration

Starting with the total concentration of butyric acid determined by GC (C_total_) and pH values along the fermentation, the non-dissociated butyric acid concentration (C_HBut_) was calculated using Henderson–Hasselbalch equation (Equation (9)), considering a pKa value (at 35 °C) of 4.83 [44].
(9)$$C_{HBut}=\frac{C_{total}}{1+10^{\left(pH-pKa\right)}}$$

## 3. Results

### 3.1. Genetic Cluster Construction Using Modular Vector pMTL83251

In order to perform the genetic construction, plasmid pMTL83251_P*_pta-ack_*_1,3-PDO_cluster was assembled by amplifying the 1,3-PDO gene cluster (5015 bp fragment on lane 3, Appendix A), which was ligated with the pMTL83251_P*_pta-ack_* backbone (4896 bp fragment on lane 2, Appendix A), resulting in the desired plasmid (9867 bp fragment on lanes 3 and 4, Appendix A). The plasmid was transformed in *E. coli* XL1-Blue MRF’. After replication of 12 different colonies of transformed *E. coli* XL1-Blue MRF’, plasmid digestion with *Pvu*II and *Xba*I confirmed that only 7 of them exhibited the correct size fragments (6445 and 3422 bp, Appendix A). One colony was selected as source of pMTL83251_P*_pta-ack_*_1,3-PDO_cluster, and its plasmid was further checked by sequencing, which also confirmed that there were no sequence mutations.

### 3.2. Tailoring a Transformation Protocol for the Br21 Strain

Initially, the protocol tested for the transformation of *C. beijerinckii* Br21 was performed as described by Little and collaborators [43]. However, after a few unsuccessful replicates, two modifications in the method were necessary, according to other authors [45,46]. The first modification was the culture used for preparing the cells’ aliquots that were stored at −80 °C. Originally, this culture had to be incubated at 37 °C until OD_600_ reached 0.2. However, in this work, the culture was incubated until OD_600_ reached 0.6, for higher cell concentration. The second modification was the amount of time necessary for cell regeneration in a fresh medium without antibiotics, immediately after the transformation. Originally, the authors incubated the cells at 37 °C for 3 h. In this work, we tested the combination of higher cell concentration with postelectroporation recovery times of 3, 4, 5, and 6 h. Subsequently, we determined that the optimal regeneration time was 5 h, while the other tested times did not result in successful transformation. After these two modifications, it was possible to transform *C. beijerinckii* Br21 via electroporation.

### 3.3. C. beijerinckii Br21 [pMTL83251_P_pta-ack__1,3-PDO_Cluster] Confirmation

After propagation of three different colonies of transformed *C. beijerinckii* Br21, their plasmids were extracted and replicated in *E. coli* XL1-Blue MRF’. The digestion of the replicated plasmids with *Pvu*II and *Xba*I confirmed that they exhibited the correct size fragments (6445 and 3422 bp, Appendix A). One of the cultures of *C. beijerinckii* Br21 [pMTL83251_P*_pta-ack_*_1,3-PDO_cluster] was further checked for strain identity, which also confirmed the Br21 strain’s successful transformation.

### 3.4. Impact of Overexpression of dhaB1, dhaB2, pduO, and dhaT in Glycerol Fermentation by C. beijerinckii Br21

Glycerol fermentation by the parent and recombinant strains was compared using a WIS medium containing 500 mmol L^−1^ glycerol. The resulting OD_600_, pH, substrate, and product variation over time for the growth experiment is illustrated in Figure 4. There was a clear difference in the performance of glycerol fermentation between *C. beijerinckii* Br21 and *C. beijerinckii* Br21 [pMTL83251_P*_pta-ack_*_1,3-PDO_cluster]. The overexpression of *dhaB1*, *dhaB2*, *pduO*, and *dhaT* increased glycerol metabolism reaction rates, where *C. beijerinckii* Br21 [pMTL83251_P*_pta-ack_*_1,3-PDO_cluster] reached stationary growth phase at around 100 h (Figure 4a) compared to the parent strain.

Overall, the transformed strain was able to grow faster compared to the parent strain, which is attested by the maximum µ (Figure 5a): 0.022 h^−1^ and 0.015 h^−1^, respectively. This is reflected in the faster pH decrease (Figure 4b), faster glycerol consumption (Figure 4c), and faster production of 1,3-PDO (Figure 4d) and butyric acid (Figure 4e). The pH decreased to 5.5 and 5.7 in the assays with *C. beijerinckii* Br21 [pMTL83251_P*_pta-ack_*_1,3-PDO_cluster] and *C. beijerinckii* Br21, respectively, due to butyric acid synthesis as sole byproduct. As seen in Figure 4e, non-dissociated butyric acid remained under 2 mmol L^−1^ for both strains. Acetate and ethanol were detected at concentrations lower than 1 mmol L^−1^ for both strains and, therefore, were not represented in Figure 4.

A decrease in pH is a possible cause for the interruption of glycerol uptake, which in this case was about 60 mmol L^−1^ for both strains. Nevertheless, the maximum q_Glycerol_ for the transformed strain was 4.5-fold higher compared to the parent strain (Figure 5b): 0.355 and 0.078 mmol glycerol mg^−1^ dry cell mass h^−1^, respectively.

The final concentration of 1,3-PDO was similar for both *C. beijerinckii* Br21 and *C. beijerinckii* Br21 [pMTL83251_P*_pta-ack_*_1,3-PDO_cluster] at approximately 35 mmol L^−1^. However, this concentration was detected after 119 h for the transformed strain, while it took 167 h for the parent strain to reach the same concentration. This results in higher 1,3-PDO productivity for *C. beijerinckii* Br21 [pMTL83251_P*_pta-ack_*_1,3-PDO_cluster] in comparison to *C. beijerinckii* Br21, 0.27 and 0.20 mmol L^−1^ h^−1^, respectively. Furthermore, the maximum q_1,3-PDO_ for the transformed strain was 1.6-fold higher compared to the parent strain (Figure 5c): 0.072 and 0.046 mmol 1,3-PDO mg^−1^ dry cell mass h^−1^, respectively. The 1,3-PDO yield at 167 h was 0.58 mol mol^−1^ for both strains, which corresponds to an efficiency of 97%, according to the theoretical stoichiometry (0.6 mol mol^−1^) (Equation (4)).

## 4. Discussion

As the exploration of *C. beijerinckii* Br21 increases in our research group, several possibilities to enhance its metabolism and industrial applicability arise. Thus, tailoring a transformation method specific for this strain was very important, as it enables the genetic manipulation of the Br21 strain by metabolic engineering and expands the number of genetic tools available for *Clostridium* genus. Moreover, further studies can benefit from the application of this synthetic biology tool.

The transformation of Gram-positive bacteria, such as *Clostridium*, is not trivial, possibly owing to their thick cell wall and high internal pressure [47]. The two modifications in the transformation protocol described by Little and collaborators [43] allowed the successful transformation of our strain. The three hours of postelectroporation recovery, intended for the proper rebuilding of the cell wall [48] described in the original protocol, were not enough for *C. beijerinckii* Br21. Extending this period for 5 h was enough, and it should not be overextended since it could result in loss of the plasmid by lack of selective pressure imposed by the antibiotic resistance gene [49]. This was observed in the transformation of *Clostridioides difficile* [49], where there was high transformation efficiency between 11 and 14 h of recovery, with a sharp decrease in efficiency at 15 h of recovery. 

Regarding the increase of cell concentration in the electroporation buffer, this also increases electroporation efficiency, up to a saturation level [48]. At this point, higher cell concentrations could lead to higher extracellular concentrations of endonucleases, which could degrade the plasmid before it enters the cell [50]. This was avoided in *C. beijerinckii* Br21 by the use of the correct methylation pattern of the plasmid.

Considering glycerol uptake, both *C. beijerinckii* Br21 and *C. beijerinckii* Br21 [pMTL83251_P*_pta-ack_*_1,3-PDO_cluster] were able to metabolize 60 mmol L^−1^, which represents only 12% of the initial glycerol added. Therefore, glycerol uptake still requires improvement, which probably could be increased by performing pH control [22] to prevent growth inhibition by low pH. In addition, culture medium optimization using the design of experiments methodology is another interesting possibility and is already in use by our research group. Using statistical tools, the relation between dependent and independent variables is evaluated through rationally designed experiments [51]. This allows us to measure the effect of dependent variables over the desired response. When applied to glycerol fermentation by *C. beijerinckii* Br21 in different culture media compositions [24], it was determined that iron sulfate had an important effect on glycerol uptake. The addition of 55 mg L^−1^ iron sulfate to the WIS medium increased glycerol consumption by 43%. 1,3-PDO final concentration was also affected, increasing by 51%. Thus, the combination of genetic engineering and culture medium optimization could further enhance 1,3-PDO synthesis by *C. beijerinckii* Br21.

The total butyric acid concentration produced by both wild and transformed strains, approximately 10 mmol L^−1^, is still under the inhibitory concentration for the Br21 strain, considering the resulting non-dissociated butyric acid. As stated in previous study from our group [22], concentrations of non-dissociated organic acids higher than 10 mmol L^−1^ can cause suppression of strain growth. In this case, the non-dissociated organic acids concentration does not seem to be the main cause of growth inhibition, since they were lower than 2 mmol L^−1^ throughout the growth experiment. One possible cause of ceased cell growth is the accumulation of 1,3-PDO along the batch fermentation assay, which could be avoided by performing fed-batch fermentation. This was accomplished in *Clostridium acetobutylicum* ATCC 824 transformed with *dhaB1*, *dhaB2*, and *dhaT* genes from *C. butyricum* VPI 3266 [52]. Under continuous fermentation conditions, the transformed *C. acetobutylicum* produced 788 mmol L^−1^ of 1,3-PDO, resulting in a yield of 0.64 mol mol^−1^ and productivity of 39 mmol L^−1^ h^−1^. Performing fed-batch fermentation led to a production of 1104 mmol L^−1^ of 1,3-PDO.

Another possible cause of cell growth inhibition is the accumulation of 3-hydroxypropionaldehyde (Figure 6), whose aldehyde group reactivity can cause bacterial DNA damage and prevent its synthesis [53]. Thus, 3-hydroxypropionaldehyde accumulation leads to the deactivation of GDHt [54]. This was determined when high intracellular concentrations of NADH were found in *C. butyricum* DSM 5431 [55], indicating that the reaction catalyzed by GDHt is the rate-limiting step in the reductive branch of glycerol fermentation. Since we overexpressed the genes related to both steps of the reductive branch, toxic intermediate accumulation still seems to have occurred. This could probably have been avoided by only overexpressing the *dhaT* gene, allowing for more 3-hydroxypropionaldehyde to be used by PDODH and lowering intracellular concentrations of NADH. However, this could also lead to limitations imposed by NADH availability. Therefore, it is undeniable that *Clostridium* metabolism is an intricate and interconnected network of reactions that must be extensively studied in order to be enhanced by synthetic biology and genetic engineering. 

In addition to its natural mutation, genetic engineering can also be used to enhance 1,3-PDO synthesis of *C. beijerinckii* Br21. Basically, four distinct strategies have been employed [57]: (1) the removal of genes related to by-product formation [58,59,60]; (2) the overexpression of genes related to the reductive branch of glycerol conversion to 1,3-PDO [61,62]; (3) the manipulation of genes related to intracellular levels of cofactors, especially NADH [9,63]; and (4) the direct evolution of enzymes from the reductive branch to improve their catalytic activity [64] or reduce coenzyme specificity [65].

Different approaches involving *dhaB*, *dhaT*, *dhaD*, and *dhaK* genes (Figure 6) were tested aiming at the increase of 1,3-PDO production from glycerol in *E. coli*, *K. pneumoniae*, and *Clostridium* spp. [8]. Zheng and collaborators [66] overexpressed *dhaB* and *dhaT* in *K. pneumoniae* DSM 2026, which did not promote an increase in 1,3-PDO synthesis. Furthermore, it delayed glycerol uptake and by-product synthesis. The authors reported that there was probably a metabolic burden due to the plasmid harboring since the logarithmic growth phase only started when GDHt and PDODH enzymatic activity decreased. This also allowed the authors to infer plasmid stability problems along the batch fermentation, which they associated with the accumulation of 3-hydroxypropionaldehyde by overexpressing *dhaB*, leading to plasmid loss. In the present study, the overexpression of *dhaB1*, *dhaB2,* and *dhaT* in *C. beijerinckii* Br21 had an opposing outcome, accelerating glycerol uptake and 1,3-PDO synthesis. Although not evaluated, plasmid loss and growth inhibition might also have taken place here imposed by 3-hydroxypropionaldehyde accumulation.

*dhaT* sole overexpression was also reported in *K. pneumoniae* KG1 [62], which did not increase 1,3-PDO production. This may be due to insufficient NADH for the PDODH reaction. Additionally, it should be noted that the limiting step of the reductive branch is catalyzed by GDHt (*dhaB*) [8], which means that the overexpression of only the *dhaT* gene does not affect 1,3-PDO synthesis but might avoid the accumulation of 3-hydroxypropionaldehyde, as previously discussed.

To overcome NADH insufficiency and lower 3-hydroxypropionaldehyde accumulation, Chen and collaborators [56] overexpressed *dhaT* and *dhaD* in *K. pneumoniae* ACCC 10082. There were increases of 3.2- and 2.6-fold in PDODH and GDHt enzymatic activities, respectively, which increased the final 1,3-PDO concentration by 56.3%.

Wischral and collaborators [9] selected GDH and DHAK as target enzymes for heterologous gene expression in *C. beijerinckii* DSM 791, which was performed using two genes from *E. coli* CA434, *gldA*, and *dhaKLM*, respectively. The CA434 strain was selected as a gene source because the corresponding enzymes generated higher titer, yield, and productivity for 1,3-PDO than those found in native 1,3-PDO producers. However, there was no increase in the final 1,3-PDO concentration by *C. beijerinckii* DSM 791. The authors suggested that, even though the oxidative branch of glycerol fermentation was overexpressed, the metabolic flux distributions remained the same. This probably can be the case in our work, using the Br21 strain. The overexpression of genes related to the reductive branch of glycerol fermentation increased GDHt and PDODH reaction rates, which probably also increased the rates of oxidative branch enzymes, in order to maintain the NADH/NAD^+^ ration. This might cause no change in metabolic flux distributions but accelerates glycerol fermentation.

The *pduO* gene was also overexpressed in *C. beijerinckii* Br21 in this work, together with *dhaB1*, *dhaB2*, and *dhaT*. This gene encodes ATP:cob(I)alamin adenosyltransferase, which catalyzes the reaction between cobalamin (vitamin B_12_) and ATP, generating adenosylcobalamin (coenzyme B_12_) and inorganic phosphate [67,68]. The GDHt produced by *K. pneumoniae* and *C. pasteurianum* is coenzyme B_12_-dependent [15], which makes the presence of *pduO* gene essential for enzyme activity. However, the GDHt produced by *C. beijerinckii* Br21 is coenzyme B_12_-independent, i.e., it does not rely on adenosylcobalamin as cofactor. Therefore, the overexpression of *pduO* gene probably did not affect the 1,3-PDO production. It might be possible that it would affect other parts of *C. beijerinckii* Br21 metabolism.

Acetate was not detected in this work, although *C. beijerinckii* Br21 is able to produce acetic acid. It is possible to assume that all the acetate produced was reassimilated by the strains, as seen in a previous work, where there was the accumulation of butyrate and consumption of acetate [26]. The excess acetoacetate generated by the reassimilation of acetate probably shifts the reaction equilibrium towards acetoacetyl-CoA formation, which subsequently leads to the production of butyrate (Figure 1). Therefore, it is likely that the concentrations of butyrate measured in our growth experiments are a result of both glycerol and acetate uptake.

As for the absence of ethanol synthesis, the overexpression of the reductive branch in glycerol fermentation might cause the use of all the reducing power for 1,3-PDO production by 1,3-propanediol dehydrogenase (Figure 1). This would lower the reaction rates towards ethanol production, explaining concentrations lower than 1 mmol L^−1^ obtained in this work. Nevertheless, the lack of ethanol can be considered advantageous since it would facilitate further product purification steps in industrial processes. Having fewer purification steps can prevent low recovery of the desired product, as well as reduce processing time and cost of production [69].

In order to gain a better understanding and interpretation of the observed metabolic profiles, several omics technologies could be employed. For instance, transcriptomics techniques such as RT-qPCR or RNA-Seq [70] could be utilized to monitor the transcription levels of key genes involved in glycerol metabolism, including *dhaB1*, *dhaB2*, *dhaT*, *dhaD*, and *dhaK*. By comparing the parent strain with the transformed strain, this approach could enable the quantification of the fold increase in expression levels and variations in expression over time. Furthermore, it could help identify underexpressed genes, potentially leading to the selection of new targets for overexpression. Finally, transcriptomics could provide insight into the regulatory mechanisms governing these key genes in response to different concentrations of initial glycerol and 1,3-PDO throughout fermentation. Another approach to consider is the utilization of metabolomics techniques, such as LC-MS or GC-MS [71], in combination with principal component analysis (PCA). This would enable the identification and quantification of the global metabolites generated during fermentation and provide a more comprehensive analysis of metabolite changes over time. Such an approach could be particularly valuable in monitoring and potentially preventing the accumulation of toxic 3-hydroxipropionaldehyde, which, as discussed previously, can lead to plasmid loss.

In summary, this study contributed to a better understanding of the metabolic modulation of glycerol conversion to 1,3-PDO by *C. beijerinckii* through genetic manipulation. Similarly, it settled a functional transformation protocol of a modular vector in this strain, which can be extended to other species of the *Clostridium* genus. The utilization of modular vectors in non-model chassis bacteria holds significant relevance due to their flexibility, standardization, scalability, and reproducibility [38]. These vectors, featuring interchangeable genetic components arranged in a standardized manner, enable researchers to rapidly construct and modify genetic constructs for specific purposes, accelerating advancements in the field [38,72,73]. In this context, our work allowed us to expand the narrow genetic toolkit available for these bacteria, which is highly relevant in biotechnological applications. Still, further research is necessary to understand the functional performance of this vector in hosts other than *C. beijerinckii*.

## 5. Conclusions

The transformation method tailored for *C. beijerinckii* Br21 was adapted from Little and collaborators [43]. After two modifications, namely in the cell concentration for transformation and the necessary time for regeneration of cells after electroporation, the protocol was successfully applicable for the Br21 strain, allowing for its transformation for the first time. Thus, overexpression of *dhaB1*, *dhaB2*, *pduO*, and *dhaT* increased 35% 1,3-PDO productivity of *C. beijerinckii* Br21 [pMTL83251_P*_pta-ack_*_1,3-PDO_cluster] compared to the parent strain. However, there was no difference in the final 1,3-PDO concentration of both strains. A combination of genetic engineering strategies and culture medium optimization for the design of experiments could probably enhance 1,3-PDO production. Additionally, fermentation mode should be considered, especially to avoid possible inhibitions caused by substrate or products.

## Figures and Tables

**Figure 1 microorganisms-11-01855-f001:**
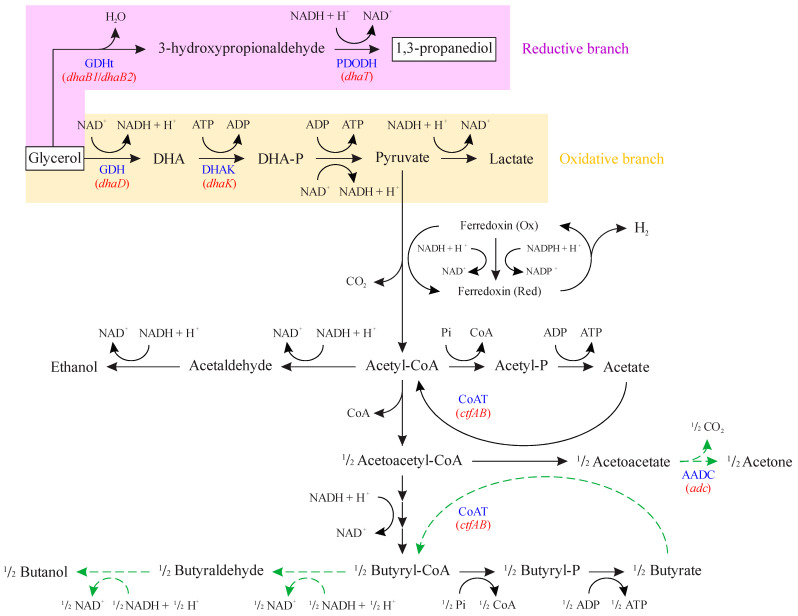
Metabolic pathways related to glycerol uptake in *Clostridium beijerinckii* Br21. DHA: dihydroxyacetone; DHA-P: dihydroxyacetone phosphate; GDH: glycerol dehydrogenase (encoded by *dhaD* gene); DHAK: dihydroxyacetone kinase (encoded by *dhaK* gene); GDHt: coenzyme B_12_-independent glycerol dehydratase (encoded by *dhaB1* and *dhaB2* genes); PDODH: 1,3-propanediol dehydrogenase (encoded by *dhaT* gene); CoAT: acetoacetyl-CoA-acetate/butyrate-CoA transferase (encoded by *ctfAB* gene); AADC: acetoacetate decarboxylase (encoded by *adc* gene). Blue font represents abbreviations for names of enzymes. Red font represents genes encoding the respective enzymes. Dashed green arrows indicate reactions that do not occur in the Br21 strain. Adapted from [16].

**Figure 2 microorganisms-11-01855-f002:**
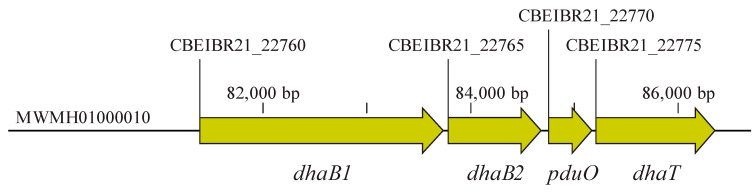
Genes related to the reductive branch of glycerol fermentation in *C. beijerinckii* Br21. These genes are clustered in Scaffold 10 (MWMH0100010, as annotated in the Nacional Center for Biotechnology Information database, NCBI), constituting a 5015-bp size fragment. *dhaB1*: gene encoding large subunit of coenzyme B_12_-independent glycerol dehydratase (locus tag: CBEIBR21_22760); *dhaB2*: gene encoding small subunit of coenzyme B_12_-independent glycerol dehydratase (locus tag: CBEIBR21_22765); *pduO*: gene encoding ATP:cob(I)alamin adenosyltransferase (locus tag: CBEIBR21_22770); *dhaT*: gene encoding 1,3-propanediol dehydrogenase (locus tag: CBEIBR21_22775).

**Figure 3 microorganisms-11-01855-f003:**
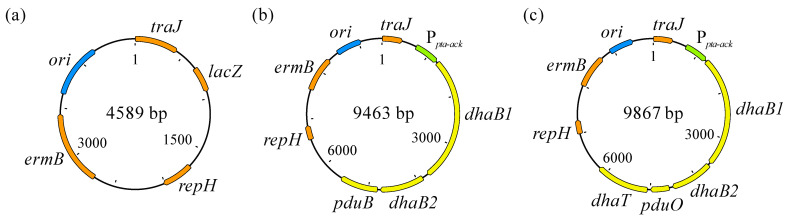
pMTL83251 (**a**), pMTL83251_P*_pta-ack_*_1,3-propanediol_CLOBI (**b**) [38], and pMTL83251_P*_pta-ack_*_1,3-PDO_cluster (**c**). *ori*: Gram-negative replicon, derived from ColE1 plasmid; *traJ*: conjugative plasmid transfer gene; *lacZ*: gene encoding β-galactosidase; *repH*: Gram-positive replicon pCB102, from *Clostridium butyricum*; *ermB*: erythromycin resistance gene; P*_pta-ack_*: phosphotransacetylase-acetate kinase genes promoter, used as constitutive promotor; *dhaB1*: gene encoding large subunit of coenzyme B_12_-independent glycerol dehydratase (locus tag: CBEIBR21_22760); *dhaB2*: gene encoding small subunit of coenzyme B_12_-independent glycerol dehydratase (locus tag: CBEIBR21_22765); *pduB*: gene encoding propanediol utilization protein; *pduO*: gene encoding ATP:cob(I)alamin adenosyltransferase (locus tag: CBEIBR21_22770); *dhaT*: gene encoding 1,3-propanediol dehydrogenase (locus tag: CBEIBR21_22775).

**Figure 4 microorganisms-11-01855-f004:**
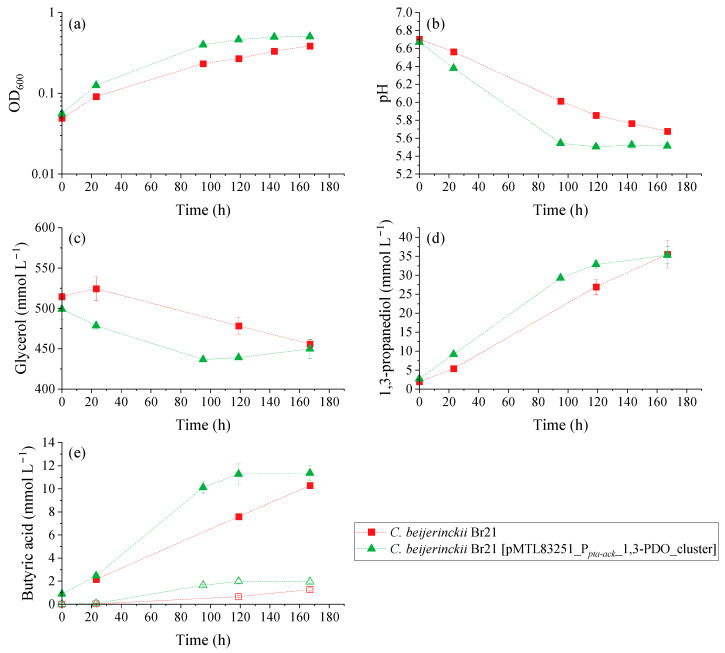
Results of the growth experiment performed with *C. beijerinckii* Br21 and *C. beijerinckii* Br21 [pMTL83251_P*_pta-ack_*_1,3-PDO_cluster] on WIS medium [40], containing 500 mmol L^−1^ glycerol. (**a**) OD_600_; (**b**) pH; and concentrations of (**c**) glycerol, (**d**) 1,3-PDO, and (**e**) butyric acid. In the butyric acid graph (**e**), solid symbols represent total acid concentration, while empty symbols represent non-dissociated acid concentration. Error bars represent standard deviations, *n* = 3.

**Figure 5 microorganisms-11-01855-f005:**
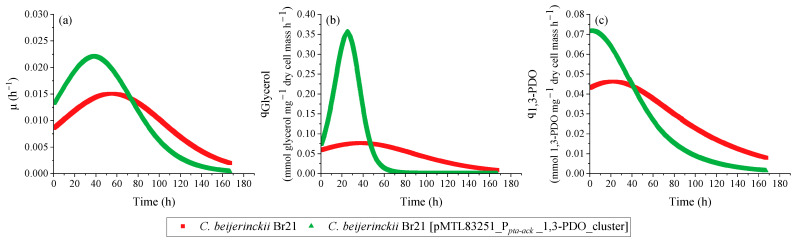
Specific rates calculated using the results from the growth experiment performed with *C. beijerinckii* Br21 and *C. beijerinckii* Br21 [pMTL83251_P*_pta-ack_*_1,3-PDO_cluster] on WIS medium [40], containing 500 mmol L^−1^ glycerol. µ (**a**), q_Glycerol_ (**b**), and q_1,3-PDO_ (**c**).

**Figure 6 microorganisms-11-01855-f006:**
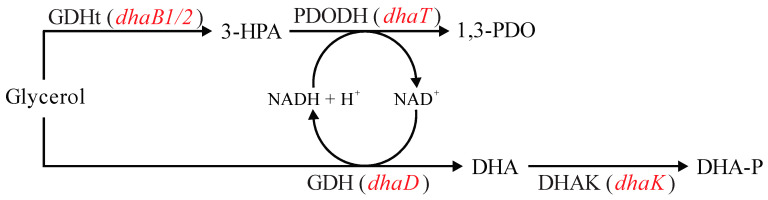
Summary of metabolic reactions related to glycerol fermentation in *Clostridium*, emphasizing the initial steps of the oxidative and reductive branches. 3-HPA: 3-hydroxypropionaldehyde; 1,3-PDO: 1,3-propanediol; DHA: dihydroxyacetone; DHA-P: dihydroxyacetone phosphate; GDHt: coenzyme B_12_-independent glycerol dehydratase (encoded by *dhaB1* and *dhaB2* genes); PDODH: 1,3-propanediol dehydrogenase (encoded by *dhaT* gene); GDH: glycerol dehydrogenase (encoded by *dhaD* gene); DHAK: dihydroxyacetone kinase (encoded by *dhaK* gene). Red font represents genes encoding the respective enzymes. Adapted from [56].

**Table 1 microorganisms-11-01855-t001:** Primers used for the amplification of the 1,3-PDO gene cluster and sequencing of pMTL83251_P*_pta-ack_*_1,3-PDO_cluster.

Name	Sequence (5′ 3′)	Application
dhaB1/2CoTdhaT.fwd	TTAAATTTAAAGGGAGGACTCTAGAATGATAAGTAAAGGATTTAGTACC	Amplification of 1,3-PDO gene cluster
dhaB1/2CoTdhaT.rev	GCAGGCTTCTTATTTTTATGCTAGCTTAATAAGCAGCTTTAAATATATTTACG	
Seq1	GGAGCTGGTGAAGTACAT	Sequencing of pMTL83251_P*_pta-ack_*_1,3-PDO_cluster
Seq2	GAAACAGAAGGTCAACCG	
Seq3	GCGTGTCAATCATTTTGG	
Seq4	AGGGAAAAGCCTTCAAGA	
Seq5	CCATCGGCATTAAAAGGT	
Seq6	GATTTTGCAGTGGAGCTT	
Seq7	TCCAAACATTCAGCCAGG	
Seq8	CCAGCAGGATTAACAGCA	
16S-27F	ATAAGCTTGGATCCAGAGTTTGATCCTGGCTCAG	Amplification of 16S rRNA
16S-1492R	ACTCGAGGATATCGGTTACCTTGTTACGACTT	

## Data Availability

All experimental data presented in this study are available upon request to the corresponding author.

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
