# Peer review of "Enhancing 1,3-Propanediol Productivity in the Non-Model Chassis Clostridium beijerinckii through Genetic Manipulation"

_microorganisms, 2023, doi:10.3390/microorganisms11071855_

Round 1
Reviewer 1 Report
In the reviewed original research article, Bortolucci et al. overexpressed the two reaction pathways for producing 1,3-propanediol (PDO) from glycerol with the non-solventogenic bacteria Clostridium beijerinckii. The article begins with a broad introduction to the topic, highlighting previous studies of the working group and the advantages of the used strain compared to other anaerobic microbes capable of producing PDO from glycerol. Furthermore, the basics of the underlying metabolic network and the current development status of metabolic engineering tools for the Clostridium genus are briefly outlined. Next, the article continues with a materials and methods section, followed by a results presentation. Here, first, the improvement of genetic tools, focusing on the transformation protocol for the Br21 strain, are given. After this, cultivation results of the original and altered Br21 strain are presented, which are thoroughly discussed in the following discussion section. The manuscript closes with a short conclusion, summarizing that genetic engineering tools were successfully adopted and optimized from previous studies and that overexpression of enzymes responsible for PDO production from glycerol (dhaB and dhaT) resulted in an improvement of 35% in the volumetric productivity of PDO production, while final concentrations remained the same. It is pointed out that further research is required, which combines metabolic engineering with media and process optimization to make use of the full potential of Br21 for PDO production from glycerol.
As a general remark, the article is professionally written, and the overall appearance is entirely satisfactory. Moreover, the work fits the journal's scope well and might be of interest to a broad readership of the journal. Nonetheless, some questions and unclarities arose while reviewing the manuscript, which should be addressed and/or answered by the authors before the article might be accepted (major revision). I hope my comments below are helpful and will contribute to improving the quality of the manuscript.
General comments:
1. In past or recent studies, other C. beijerinckii strains have also been manipulated to produce PDO from glycerol (see doi: 10.3390/microorganisms11030784 or doi: 10.1016/j.biortech.2016.04.020). What makes the present study in the reviewed manuscript novel and unique in this context? Is it only the use of the Br21 strain in contrast to the DSMZ strain, or can the authors raise any further points that make the reviewed manuscript impactful? Please comment and complement the manuscript accordingly.
2. Diallo et al. (doi: 10.1016/j.ymeth.2019.07.022) developed a much-improved transformation approach for C. beijerinckii. Why has this approach not been pursued in this study? Please explain.
3. The cross-references for figures seem to contain errors and cause issues with formatting. Please correct this for the revised version.
4. The manuscript generally lacks quantitative evidence to explain and justify the claimed different behaviors of the newly generated and original Br21 strain. This is also addressed in the discussion, pointing out that Clostridia metabolism is highly complex, and it is exceptionally challenging to understand the different metabolic behaviour. Since it is expected that the focus of this work is clearly on the genetic engineering part, it is acceptable that not all of the observed phenomena might be explained in the framework of this study. Anyhow, it would be much appreciated if the authors would name future strategies in more detail that would enable them to unveil more of the observed metabolic patterns and help in the future development of the process. Examples for future work would be metabolomic studies (which metabolites should be quantified?), fluxomics (how do the fluxes change quantitatively?) or transcriptomics. Please add some more details in this regard to the manuscript and do not only name methods but particular targets and promising methods.
Specific comments:
1. Lines 43 and the following: Using (raw) glycerol as a promising substrate for bioprocesses is highlighted. Is glycerol indeed a future-proof substrate for the bioeconomy? In this context, please add current numbers on its availability and future projections for this feedstock.
2. Lines 71 and the following: it is written that the used Br21 strain lacks the adc gene, resulting in the strain being “incapable of acetone, butanol, ethanol (ABE) fermentation”. Please add “conventional” (or a similar word) before ABE. Otherwise, the reader might misunderstand the statement in a way that ethanol and/or butanol production might not be feasible at all, only by the deletion/missing of the adc gene.
3. Equation 1: this equation presents the overall reaction stochiometry at maximal PDO production. Do metabolic reducing equivalent ((NAD(P)+/NAD(P)H) and energy balances (ADP/ATP) also close in this scenario? Please comment.
4. Figure 1: Please check the arrows for the LDH (NADH should be used, not produced). Furthermore, please add co-factors and correct stoichiometry downstream of acetyl-CoA. For example, 2 mol of acetyl-CoA should be required to produce 1 mol acetoacetyl-CoA. What about reducing equivalents in the conversion of acetyl-CoA down to butyryl-CoA? Please amend the figure accordingly.
5. Line 124: How many µL of the cryo culture were used for the initial starting culture of 10 mL? Please add this information to the manuscript.
6. Line 130: It is said that anaerobic conditions were established “by gas phase exchange“ – please specify and describe the practical procedure in more detail.
7. Line 227: The authors write, "All the procedures were done under anaerobic conditions“. How were those achieved? Were they performed in an anaerobic chamber? Please explain and add the information to the manuscript.
8. Line 296: Why was HCl added to the supernatant? Please comment.
9. Line 372: It is written that “Overall, the transformed strain was able to grow faster compared to the parent strain“. The figure slightly indicates this, but please prove this by stating quantitative evidence. Calculate the growth rate from the first two data points and state/compare them.
10. Line 384: Here, it is written that “The pH decrease causes interruption of glycerol uptake“ – how was this concluded to be the primary reason for the decrease of glycerol uptake? Please comment and complement the manuscript accordingly.
11. Lines 385 and the following: Volumetric productivities and yields are presented and discussed. Here, it is said that the generated strain offered improved volumetric productivity after 119 h. However, the PDO yields of the generated and original BR21 strains seem to have been compared only for the last data points. How were the compared yields after 119 h? What about the estimated cell-specific substrate uptake and production rates? Can it indeed be concluded from the data that the altered strain outperformed the original Br21 strain? Please comment and add further information and numbers to the manuscript.
12. Figure 5: Only NADH formation and use in the oxidative and reductive branches are considered here. What about NADH use in the butyrate formation pathway? Why is this not considered here and in the discussion around the figure? Please comment and change the manuscript when required.
Author Response
Dear Reviewers
We are submitting a revised version of the manuscript entitled "Enhancing 1,3-propanediol productivity in the non-model chassis Clostridium beijerinckii through genetic manipulation" along with our responses to the reviewers' comments and the suggested amendments.
We sincerely appreciate the valuable suggestions provided by the reviewers, which have significantly enhanced the quality of our manuscript. We believe that the implemented amendments align with the recommendations and are comprehensive enough to meet the publication standards of Microorganisms.
Thank you for your time and consideration. We look forward to your favorable response.
Best regards,
Valeria Reginatto

Reviewer 2 Report
In this study, authors overexpressed the native 1,3-PDO pathway genes, i.e., dhaB1, dhaB2, dhaT, and pduO, in the Clostridium beijerinckii Br21 strain, and investigated the impact on cell growth, glycerol consumption, 1,3-PDO production, byproducts (such as butyrate, acetate and ethanol) formation, and the culture medium pH over the time course of a batch fermentation process. The authors found that overexpression of the 1,3-PDO gene cluster improved the PDO productivity while the titer was approximately the same as that of the parental strain after 7 days. It was postulated that drop of culture pH was possibly the cause of decreased cellular activity. The study also improved a previously reported transformation method. While this study has provided some interesting findings in genetic engineering of Clostridium beijerinckii for enhanced production of 1,3-PDO, I suggest the authors address the following concerns to improve the clarity and quality of the manuscript.
Major concerns:
1) The engineered strain exhibited enhanced glycerol consumption, cell growth, and PDO productivity during the 0-120 h period, whereas the cellular metabolism slowed down hereafter as the pH of the culture decreased. The culture pH seems playing a major role in modulating the metabolism in Clostridium beijerinckii, and the authors mentioned that similar findings were reported in previous studies. Adjusting pH could be a very simple experiment to run, e.g., by adding calcium carbonate, or manually adjust the culture medium pH using NaOH etc. By doing so, the authors might be able to see more sustained enhanced 1,3-PDO productivity in the engineered strain.
2) The authors claimed that a modified transformation protocol was established so that the transformation efficiency in C. beijerinckii was improved. While the transformation method was indeed improved by the authors’ efforts, a more systematic optimization of the transformation protocol, e.g., harvesting cells at series of varied OD, and a series of postelectroporation recovery time, etc, would make the optimization more convincing and the method more impactful.
Minor concerns:
1) Line 59, “dihydroxyacetone kinase (GDH, encoded by dhaD gene)” should be “glycerol dehydrogenase (GDH,...”
2) The authors attributed the low level accumulation of acetate to readily re-assimilation of acetate, but if that’s the case there might be significant accumulation of acetoacetate (Figure 1). Have the authors been able to detect it by HPLC?
3) The authors discussed the function of pduO gene in line 525-534. Since the authors already constructed a recombinant plasmid overexpressing dhaB1, dhaB2 and dhaT, but not pduO (line 175-176; Figure 3), why the authors did not include that strain in the study to rule out the essentiality/role of the pduO in PDO production in the C. beijerinckii Br21 strain?
Generally acceptable. Minor revisions are needed.
Author Response

(The authors gave the same response as above.)

Round 2
Reviewer 1 Report
Significant amendments have been made to the manuscript, improving the quality of the article. In particular, the addition of section 2.7 and figure 5 are much appreciated. The article might now be accepted for publication in its current form.
Author Response
Thank you for your insightful suggestions. They have certainly improved our manuscript.
Reviewer 2 Report
I agree to accept the revised manuscript for publication.
Author Response

(The authors gave the same response as above.)
